# Optical Properties in a ZnS/CdS/ZnS Core/Shell/Shell Spherical Quantum Dot: Electric and Magnetic Field and Donor Impurity Effects

**DOI:** 10.3390/nano13030550

**Published:** 2023-01-29

**Authors:** Rafael G. Toscano-Negrette, José C. León-González, Juan A. Vinasco, A. L. Morales, Fatih Koc, Ahmet Emre Kavruk, Mehmet Sahin, M. E. Mora-Ramos, José Sierra-Ortega, J. C. Martínez-Orozco, R. L. Restrepo, C. A. Duque

**Affiliations:** 1Grupo de Materia Condensada-UdeA, Instituto de Física, Facultad de Ciencias Exactas y Naturales, Universidad de Antioquia UdeA, Calle 70 No. 52-21, Medell AA 1226, Colombia; 2Departamento de Física y Electrónica, Universidad de Córdoba, Carrera 6 No. 77-305, Montería 230002, Colombia; 3Department of Metallurgical and Materials Engineering, Ahi Evran University, Kirsehir 40000, Turkey; 4Physics Department, Faculty of Sciences, Selcuk University, Konya 42075, Turkey; 5Department of Nanotechnology Engineering, Abdullah Gul University, Kayseri 38080, Turkey; 6Centro de Investigación en Ciencias, Instituto de Investigación en Ciencias Básicas y Aplicadas, Universidad Autónoma del Estado de Morelos, Av. Universidad 1001, Cuernavaca CP 62209, Morelos, Mexico; 7Grupo de Investigación en Teoría de la Materia Condensada, Universidad del Magdalena, Santa Marta 470004, Colombia; 8Unidad Académica de Física, Universidad Autónoma de Zacatecas, Calzada Solidaridad Esquina con Paseo La Bufa S/N., Zac., Zacatecas CP 98060, Mexico; 9Universidad EIA, Envigado CP 055428, Colombia

**Keywords:** core/shell/shell quantum dot, donor impurity, external electric and magnetic field, absorption coefficients

## Abstract

A theoretical analysis of optical properties in a ZnS/CdS/ZnS core/shell/shell spherical quantum dot was carried out within the effective mass approximation. The corresponding Schrödinger equation was solved using the finite element method via the 2D axis-symmetric module of COMSOL-Multiphysics software. Calculations included variations of internal dot radius, the application of electric and magnetic fields (both oriented along *z*-direction), as well as the presence of on-center donor impurity. Reported optical properties are the absorption and relative refractive index change coefficients. These quantities are related to transitions between the ground and first excited states, with linearly polarized incident radiation along the *z*-axis. It is found that transition energy decreases with the growth of internal radius, thus causing the red-shift of resonant peaks. The same happens when the external magnetic field increases. When the strength of applied electric field is increased, the opposite effect is observed, since there is a blue-shift of resonances. However, dipole matrix moments decrease drastically with the increase of the electric field, leading to a reduction in amplitude of optical responses. At the moment impurity effects are activated, a decrease in the value of the energies is noted, significantly affecting the ground state, which is more evident for small internal radius. This is reflected in an increase in transition energies.

## 1. Introduction

The study of low-dimensional semiconductor heterostructures (LDSH) has gained relevance due to their high-efficient optoelectronic properties. In accordance, numerous theoretical and experimental studies have been performed [1,2,3,4,5]. Among these structures it is possible to mention the quantum wells (with charge carrier confinement along one dimension), quantum wires (confinement in two dimensions), and quantum dots (QDs). In the latter, electrons (holes) are confined in all spatial directions, allowing for electronic properties of great scientific and technological interest. Such features can be controlled by modifying the size and/or the geometry of the system through the application of external electromagnetic probes, hydrostatic pressure, and temperature; and also by suitable doping with impurity atoms [6,7,8,9,10,11,12,13,14,15]. These LDSH have found practical realization in nanophotonics and nanoelectronics thanks to controlled fabrication using techniques such as molecular beam epitaxy (MBE), ultrasonic sol-gel, and hydrothermal method, among others; with nanostructure sizes below 10 nm [16,17,18,19,20,21].

Among the most studied QDs, different geometrical shapes are present: cylindrical, pyramidal, conical, and spherical [22,23,24,25]. In this work, we focus on studying a kind of core/shell/shell spherical QD, which consists of a semiconductor material surrounded by another one which exhibits a different value of the energy band gap. In a QD, discrete energies depend on size, external fields, band gaps, etc., with which physical and optical properties can be controlled and used for applications in fields such as electronics, medicine, chemistry, and biology [26]. The most used semiconductor compounds to fabricate QDs are those formed by atoms of II–VI, III–V, and IV–VI groups of the periodic table. Lately, researchers have become of significant interest in studying core-shell QDs based on ZnS/CdS semiconductors, which find a number of applications in photovoltaics, optoelectronics, medicine, and so on [27,28,29,30]. For example, Linkov et al. demonstrated that multicomponent core/shell CdSe/ZnS/CdS/ZnS QDs can be used as optimal fluorescent probes for the development of systems for cancer diagnosis and treatment using anticancer compounds based on acridine derivatives [31]. In 2019, Zeiri et al. conducted a study on the third-order nonlinear optical susceptibility for CdS/ZnS/CdS/ZnS core/shell/well/shell spherical QDs, reporting that both the position and the intensity of the peaks can be controlled by varying the thickness of the shell. Furthermore, as the width of the inner or outer layer increases, the susceptibility peaks are shifted towards the red with increasing intensities [32]. In 2022, Kuzyk et al. studied the possible applications in nanomedicine of deformation effects in CdSe-core/ZnS/CdS/ZnS-shell QDs, and how they interact with human serum albumin, finding that QDs with three layers are more sensitive to deformation and, at the same time, the strain is almost independent of the radius of the core. In the proposed system, significant strains arise in the CdSe/ZnS/CdS/ZnS/Qd-human serum albumin bionanocomplexes, which can lead to an energy shift of the conduction band edge by 40 meV. Lastly, they state that their results can be used to develop bionanosensors for the determination of albumin concentration [33].

Here, we shall use the effective mass approximation to conduct a theoretical analysis of ZnS/CdS/ZnS core-shell-shell QD optical properties. The electronic states in the system are numerically determined using finite element method (FEM) as implemented in COMSOL-Multiphysics package in the axis-symmetric module. Calculations include the variation of internal radius ($R_{1}$), as well as the change in the intensities of external magnetic and electric fields, with and without the effects of on-center donor impurity. The analysis of optical response goes through the evaluation of linear and nonlinear coefficients of both light absorption and relative refractive index change from the corresponding expressions derived within the density matrix formalism. The article’s organization is as follows: Section 2 presents the theoretical framework, in Section 3, we discuss the results, and Section 4 is devoted to the main conclusions of the work.

## 2. Theoretical Model

Figure 1 shows the investigated system, a ZnS/CdS/ZnS spherical QD of the core/ shell/shell type (a). The dimensions of the core, inner shell, and outer shell are $R_{1}$, $R_{2}$, and $R_{3}$, respectively (b). In (b), the lowest part of the figure shows a schematic view of the radial-dependent confinement potential. This function is set as zero within the CdS material, $V_{0}$ in the ZnS material, and infinite in the vacuum-external region. Effects of an on-center donor impurity, an axially applied magnetic ($\vec{B}$), and an electric (${\vec{F}}_{z}$) field, have been taken into account. In Figure 1c, the projection on the plane φ=0 of the structure is presented. The rotation of the region shown around the *z*-axis gives rise to the structure shown in Figure 1a. In addition, the particular mesh used in FEM calculations is schematically presented. It is readily apparent that in the region of the first shell layer, much greater refinement of the mesh has been considered compared to that chosen for the other two regions. Dirichlet boundary conditions are assumed on the semicircle of radius $r=R_{3}$ (the outer one).

In Cartesian coordinates, the parabolic band effective-mass Schrödinger equation for a conduction electron confined in the structure, with all the contributions mentioned above, is written in the form:(1)$$\left[\frac{1}{2\;m^{*c}}{\left(\vec{p}+e\;\vec{A}\right)}^{2}+e\;F_{z}\;z+V\left(x,y,z\right)-\frac{\kappa \;e^{2}}{4\;\pi \;\varepsilon_{0}\;\varepsilon_{r}^{c}\;R}\right]\psi \left(x,y,z\right)=E\;\psi \left(x,y,z\right)\;,$$
where $m^{*c}$ is the electron effective mass, $\varepsilon_{r}^{c}$ is the static dielectric constant (the index *c* indicates the core or shell materials), $R=\sqrt{x^{2}+y^{2}+z^{2}}$ is the electron-impurity distance (with the shallow-donor impurity placed at the center of the structure), $\vec{p}=-i\;\hbar \;\vec{\nabla }$, *e* is the absolute value of the electron charge, and V(x,y,z)=V(r) (with V(r) expressed in cylindrical coordinates) is the radially symmetric confinement potential. Here, κ is a parameter that controls the presence or absence of the shallow donor impurity (κ=0 removes the impurity effects whereas κ=1 turns them on). Besides, $\vec{A}=-\frac{B}{2}\;\left(y\;\hat{i}-x\;\hat{j}\right)$ represents the potential vector associated to the applied magnetic field, where $\vec{B}=\vec{\nabla }\times \vec{A}$ comes from the symmetric gauge. As mentioned, we impose Dirichlet boundary conditions at the outer edges of the barrier matrix. The process also assumes BenDaniel-Duke conditions at the inner QD interfaces (see Figure 1b,c).

The $m^{*c}$ and V(r) terms in Equation (Equation 1) depend on the radial position in the heterostructure and they are expressed as:(2)$$m^{*c}=\left\{\begin{array}{cc} m^{*\mathrm{ZnS}}\;, & \mathrm{if}\;0\leq r\leq R_{1}\;,\\ m^{*\mathrm{CdS}}\;, & \mathrm{if}\;R_{1}<r\leq R_{2}\;,\\ m^{*\mathrm{ZnS}}\;, & \mathrm{if}\;R_{2}<r\leq R_{3}\;, \end{array}\right.$$
and
(3)$$V\left(r\right)=\left\{\begin{array}{cc} V_{0}\;, & \mathrm{if}\;0<r\leq R_{1}\;,\\ 0\;, & \mathrm{if}\;R_{1}<r\leq R_{2}\;,\\ V_{0}\;, & \mathrm{if}\;R_{2}<r\leq R_{3}\;,\\ \infty \;, & \mathrm{if}\;r>R_{3}\;. \end{array}\right.$$

Expanding Equation (Equation 1), and using the azimuthal symmetry condition of the system, it is possible to propose in cylindrical coordinates a solution of the type $\psi \left(x,y,z\right)=\psi \left(r,\varphi ,z\right)=R\left(r,z\right)\;e^{i\;m\;\varphi }$, where the *m* is the principal quantum number. Consequently, the R(r,z) function satisfies the differential equation
(4)$$\begin{array}{cc} \; & \left[-\frac{\hbar^{2}}{2\;m^{*c}}\nabla_{r,z}^{2}+\frac{m^{2}\;\hbar^{2}}{2\;m^{*c}\;r^{2}}+\frac{\hbar \;e\;B\;m}{2\;m^{*c}}+\frac{e^{2}\;B^{2}\;r^{2}}{8\;m^{*c}}+e\;F_{z}\;z\right.\\ \; & \left.-\frac{k\;e^{2}}{4\;\pi \;\varepsilon_{0}\;\varepsilon_{r}^{c}\;R}+V\left(r\right)\right]R\left(r,z\right)=E_{m}\;R\left(r,z\right), \end{array}$$
where $\nabla_{r,z}^{2}$ is the *r*- and *z*-dependent two-dimensional Laplacian operator and $r=\sqrt{r^{2}+z^{2}}$ is the electron-impurity distance.

The study is focused on electron confinement effects related to QD dimensions and on the influence of externally applied electric and magnetic fields, with and without on-center donor impurity effects. The Equation (Equation 4), is solved via the FEM [34,35,36], using the COMSOL-Multiphysics licensed software [37,38,39], with which it is possible to obtain the wave functions and the energies of the different states. The settings used to build an extra fine, user-controlled mesh are: 6185 mesh vertices, 12,104 triangles, 460 edge elements, 10 vertex elements, 0.6179 minimum element quality, 0.8964 medium element quality, 0.2051 element area ratio, and 226.2 nm${}^{2}$ mesh area. Employed hardware contains a single 11th-generation i7 processor. Obtained electron states are then used to evaluate the optical properties (absorption and the relative refractive index change coefficients) associated with transitions between the lowest two energy levels (the ground state and the first excited state). Keeping in mind the characteristic spherical symmetry of the system, the expression to evaluate the first-order absorption coefficients is:(5)$$\begin{array}{c} \alpha^{\left(1\right)}\left(\omega \right)=\sqrt{\frac{\mu_{0}}{\varepsilon_{r}\;\varepsilon_{0}}}\;\;\frac{\omega \;e^{2}\;\sigma \;\hbar \;\Gamma_{12}\;{\left|M_{12}\right|}^{2}}{{\left(E_{12}-\hbar \;\omega \right)}^{2}+{\left(\hbar \;\Gamma_{12}\right)}^{2}}\;. \end{array}$$

The expression to evaluate the third-order absorption coefficients is given by
(6)$$\alpha^{\left(3\right)}\left(\omega ,I\right)\;=\;-\;\sqrt{\frac{\mu_{0}}{\varepsilon_{r}\;\varepsilon_{0}}}\;\;\frac{2\;I}{n\;\varepsilon_{0}\;c}\;\;\frac{\omega \;e^{4}\;\sigma \;\hbar \;\Gamma_{12}\;{\left|M_{12}\right|}^{4}}{{\left[{\left(E_{12}-\hbar \;\omega \right)}^{2}+{\left(\hbar \;\Gamma_{12}\right)}^{2}\right]}^{2}}\;.$$

The total absorption coefficient is the sum of the linear and the third order:(7)$$\begin{array}{c} \alpha \left(\omega ,I\right)=\alpha^{\left(1\right)}\left(\omega \right)+\alpha^{\left(3\right)}\left(\omega ,I\right)\;. \end{array}$$

The corresponding expressions to evaluate the relative refractive index change are, respectively:(8)$$\frac{\Delta n^{\left(1\right)}\left(\omega \right)}{n}=\frac{\sigma \;e^{2}\;{\left|M_{12}\right|}^{2}}{2\;\varepsilon_{0}\;\varepsilon_{r}}\;\;\frac{E_{12}-\hbar \;\omega }{{\left(E_{12}-\hbar \;\omega \right)}^{2}+{\left(\hbar \;\Gamma_{12}\right)}^{2}}\;,$$
(9)$$\frac{\Delta n^{\left(3\right)}\left(\omega \right)}{n}=\frac{-\sigma \;\mu_{0}\;c\;I}{n^{3}\;\varepsilon_{0}}\;\frac{e^{4}\;{\left|M_{12}\right|}^{4}\;\left(E_{12}-\hbar \;\omega \right)}{{\left[{\left(E_{12}-\hbar \;\omega \right)}^{2}+{\left(\hbar \;\Gamma_{12}\right)}^{2}\right]}^{2}}\;,$$
and
(10)$$\frac{\Delta n(\omega ,I)}{n}=\frac{\Delta n^{\left(1\right)}\left(\omega \right)}{n}+\frac{\Delta n^{\left(3\right)}\left(\omega ,I\right)}{n}\;.$$

The above expressions arise from the density matrix formalism as developed in Ref. [40], where $E_{12}=E_{2}-E_{1}$ is the transition energy between the initial (“1”) and final (“2”) state. In addition, $M_{12}$ represents the reduced off-diagonal dipole matrix element between the two electronic states (divided by the charge of the electron). For *z*-polarized incident radiation, this quantity can be written as $M_{12}=\langle \psi_{1}\left(r,\varphi ,z\right)|z|\psi_{2}\left(r,\varphi ,z\right)\rangle$. Additionally, $\hbar \;\Gamma_{12}$ is a damping term associated with the electron lifetime due to the dispersion between subbands (which, here, has a value of 0.7 meV). Besides, $n=\sqrt{\varepsilon_{r}}$ is the relative refractive index, $\mu_{0}$ is the vacuum magnetic permeability ($4\;\pi \times 10^{-7}$ T m A${}^{-1}$), σ is the charge carrier density, set at $3.0\times 10^{22}$ m${}^{-3}$. Finally, *I* represents the intensity of the optical radiation in the system, whose value has been chosen equal to 30 MW/m${}^{2}$.

The interaction energy between the electron and the impurity is known as the binding energy. It is calculated from the following expression:(11)$$E_{b}=E_{\kappa =0}-E_{\kappa =1},$$
where $E_{\kappa =0}$ and $E_{\kappa =1}$ are the ground state energies without (κ=0) and with impurity (κ=1), respectively.

## 3. Results and Discussion

Besides values above commented for some input parameters, the remaining ones used in this work are: $m^{*\mathrm{CdS}}=0.19\;m_{0}$, $m^{*\mathrm{ZnS}}=0.25\;m_{0}$, $\varepsilon^{\mathrm{CdS}}=8.3$, $\varepsilon^{\mathrm{ZnS}}=8.9$, and $V_{0}=0.8$ eV. Here, $m_{0}$ is the free electron mass [41]. In Equations (Equation 5)–(Equation 10), the parameters *n* and $\varepsilon_{r}$ are those corresponding to CdS.

In Figure 2, the lowest electronic states of a ZnS/CdS/ZnS spherical core/shell/shell QD, for $F_{z}=0$ and B=0, are depicted as functions of the $R_{1}$ radius, with and without shallow-donor impurity effects. Figure 2a,b show that the energy of the three lowest levels increases as $R_{1}$ augments. This effect is associated with the Heisenberg uncertainty principle since as the inner radius becomes larger, the thickness of the CdS shell, where the electron is confined, decreases. Greater confinement produces an increase in energy. Figure 2b shows the effect of a shallow donor impurity placed at the QD center. In this case, the energy levels shift toward lower energies, below the conduction band, due to attractive electron-impurity interaction. The minimum value of the electron-impurity distance is for the ground state (m=0). This explains why this state is the one that undergoes a greater displacement towards lower energies. The inset in Figure 2a shows the electron-impurity binding energy as a function of the internal radius. One may notice a decrease in the binding energy as the internal radius grows. For example, for $R_{1}=1$ nm, it has a value of 39.195 meV, while for $R_{1}=4$ nm is 24.392 meV. In fact, as the radius $R_{1}$ increases, the electron moves further away from the impurity, and this causes the impurity-electron Coulomb interaction to decrease. Due to the spherical symmetry of the QD, in the absence and presence of impurities at the center of the quantum dot, the energies shown in Figure 2a,b must have the same degeneracies of a bulk hydrogen atom. For example, the first excited state must be triply degenerate, with the three solutions m=0,±1. Then, the second excited state has a five-fold degeneracy: m=0,±1,±2.

With the aim of confirming the results obtained, we have proceeded to derive the solution directly from Equation (Equation 1), which corresponds to a three-dimensional problem. The corresponding outcome appears plotted in Figure 2 using solid symbols. As can be seen, the results exactly coincide for both the 3D solution, Equation (Equation 1), and the 2D solution, Equation (Equation 4), which uses the axial symmetry of the problem. Likewise, in the case of the configuration without impurity, in the absence of applied fields, the problem of an electron confined in a multilayer QD with spherical symmetry has a quasi-exact solution. Accordingly, the wave function in the different regions of the QD can be written as a linear combination of radial Bessel functions. Applying the Ben Daniel-Duke boundary conditions, we arrive at a transcendental equation whose numerical solution leads to obtaining the energies of the confined particle. The full symbols have also been obtained by solving this quasi-exact problem, again showing an excellent agreement for the three calculation procedures used. One of the differences between obtaining the solution of Equations (1) and (4) lies in the computation time. Using a 2.8 GHz Intel i7-processor, obtaining the ground state energy [using Equation (Equation 4)], for a single value of $R_{1}$ in Figure 2a, takes 3 s of computing time, a situation that increases by a factor of 4 when calculating the same energy in the 3D-module that corresponds to the solution of Equation (Equation 1).

Figure 3 contains plots of the lowest three electronic states in a ZnS/CdS/ZnS spherical core/shell/shell QD as functions of the applied magnetic field with and without on-center shallow donor impurity effects for several values of the *m*-quantum number. From Figure 3a, it is possible to observe that the magnetic field breaks the first excited state triple degeneracy and the second excited state degeneracy. States with a positive value of *m* always increase in energy with the strengthening of the applied magnetic field. For states with a negative *m*-value, energies initially show a decreasing character with the magnetic field. They evolve to reach a minimum value, becoming the actual ground state of the system and, then, start to augment. Consequently, as a result of the modification of the energy spectrum due to the increment in magnetic field intensity, the ground state does not preserve the *s*-like symmetry. From this figure, it is also noticed that, with a zero magnetic field, the first and second excited states have degeneracies of order three and five, respectively, as indicated in the discussions of Figure 2a,b. On the other hand, Figure 3b shows the energies as functions of the external magnetic field, taking into account the effects of the donor impurity at the center of the QD. As commented, energy levels shift towards lower values due to the attractive nature of the Coulombic coupling with the ionized impurity. It is seen that in Figure 3a,b, the energy behavior shows the same trend. Actually, the impurity effect under the magnetic field is not very noticeable compared to the system without the impurity since it changes from about 3.723 meV to 3.874 meV. This is reflected in the binding energy (see inset in Figure 3b), where it increases as the magnetic field increases.

Results for allowed electron energies as functions of an externally applied static electric field appear plotted in Figure 4, without and with on-center donor impurity effects for B=0. Figure 4a shows that energy drops as the intensity of the applied electric field increases. Due to the spherical symmetry of the system, the same behavior for negative values of the electric field occurs, so the plot only shows positive values of field intensity. In the high electric field regime, the wave function is compressed towards the regions with $z=-R_{2}$, the region at which the confining potential becomes deeper due to the negative potential contribution. As observed, the electric field breaks the triple degeneracy of the first excited state, giving rise to a doubly degenerate energy state that comes from the solutions with m=±1 and to a non-degenerate state for m=0; for which the two antinodes are located along the *z*-axis. This state responds differently to the electric field than the states with m=±1, since it initially shows a growing behavior, after which there begins a descent when the field becomes strong enough.

Figure 4b presents the effects of the donor impurity combined with the electric field. The first thing to highlight is that there is a drop in the energy values compared to the situation depicted in Figure 4a. The second aspect to note is the systematic decrease of the energies as the electric field increases—opposite to the trend reported in Figure 3b—and the presence of the same kind of degeneracies discussed above. This has to do with the impurity position at the center in the QD, which preserves the spherical symmetry. When analyzing the behavior of binding energy as the electric field increases (see inset in Figure 4b), it is noticed a reduction in its value; for example, for a zero electric field, we have a binding energy of 24.392 meV, and for an electric field of 50 kV/cm is 23.443 meV. The electric field confines the electron to the inner edge of the region $z=-R_{2}$, which implies a drop in the Coulomb interaction between the impurity and the electron since the effective electron-impurity distance augments.

Now, we report on the behavior of squared reduced dipole moment matrix element ($|M_{12}{|}^{2}$) for electron transitions from the ground state (|1〉) to the first excited state (|2〉) in a ZnS/CdS/ZnS spherical core/shell/shell QD, with m=0. Results will be presented taking into account the influence of external fields—electric and magnetic—oriented along the *z*-direction, as well as the changes in the QD dimensions, controlled through $R_{1}$. Analysis of Figure 5, Figure 6 and Figure 7 requires an understanding of wave functions symmetries. Transitions studied are those involving states with m=0 since the used incident radiation is linearly polarized in the *z*-direction, thus making the transitions to states with m≠0 to be forbidden.

Variation of $|M_{12}{|}^{2}$ and transition energy, $E_{12}$, as functions of the internal radius for a confined electron in a ZnS/CdS/ZnS spherical QD with and without the effects of on-center donor impurity is shown in Figure 5. According to Figure 5a, by increasing the QD internal radius, the size of the shell -where the electron is confined- decreases, causing a linear increase of $|M_{12}{|}^{2}$. This behavior is shown in the inset, from which one may conclude that the spatial overlap between wave functions increases when moving from 1 nm to 8 nm in the internal radius. In Figure 5b, with the inclusion of donor impurity influence, a similar behavior is observed for $|M_{12}{|}^{2}$. From the inset in Figure 5b, it is seen that the ground state wave function is attracted to the impurity while the first excited wave function remains almost unchanged. Hence, the overlap between both states decreases concerning the case without impurity. For example, $|M_{12}{|}^{2}$ has a value of 13.330 nm${}^{2}$ for $R_{1}=1$ nm without impurity. With impurity effects, the value falls to 8.380 nm${}^{2}$, with a difference of 4.950 nm${}^{2}$, which becomes smaller as the radius increases, weakening the Coulombic interaction. Figure 5c,d show the corresponding variation of the transition energy from the ground state to the first excited state, with and without impurity, from which it is seen that as the $R_{1}$ increases, this energy decreases. As the internal radius increases, the shell decreases, causing the energies of the ground state and the first excited state to increase, becoming closer to each other for large radii (see Figure 2). From these two plots, Figure 5c,d, one may conclude that the transition energy is higher for small internal radii with donor impurity effects. This happens because, at a small internal radius, the interaction between the impurity and the electron is greater, causing the electron energies to fall below the conduction band (see Figure 2b). But, when the inner radius gets larger, the impurity-electron interaction becomes weaker, making the transition energy similar to the case without impurity. For example, when $R_{1}=6$ nm, without impurity, the transition energy is 5.820 meV, and with impurity, it is 5.876 meV, having a difference of 0.056 meV.

Plots in Figure 6, represent results for the same quantities as in Figure 5 but considering the variation of the intensity of the applied static magnetic field. From Figure 6a, it is noticed an increase in $|M_{12}{|}^{2}$ as the strength of the applied magnetic field increases. This happens because the ground state gradually changes from a uniformly distributed probability density at B=0 to a concentrated probability density at the upper and lower regions of the QD for B=30 T. At these regions, the first excited state presents its maximum probability density, so the overlap between wavefunctions becomes larger. Effects associated with an on-center donor impurity appear in Figure 6b, showing the same behavior as in Figure 6a. That is, when comparing the values of $|M_{12}{|}^{2}$ for the system with and without impurity, a lower value is seen when there are impurity effects. For example, when B=0, in the absence of any impurity, $|M_{12}{|}^{2}$ has a value of 19.024 nm${}^{2}$. In the case that considers the presence of an ionized impurity, it has a value of 18.208 nm${}^{2}$. Switching on the magnetic field to B=30T leads to a similar behavior. In the case without impurities, it has a value of 21.760 nm${}^{2}$, and for the case with an on-center donor, it has a value of 20.581 nm${}^{2}$. This is because, at zero applied magnetic field, the ground state wave function has its probability density concentrated around the impurity. The first excited state presents a more significant probability density localization at the upper and lower QD regions, thus producing a smaller overlap between the involved states. In the 30 T case, the ground state wave function probability density is concentrated at the upper and lower areas of the QD. Consequently, the overlap with the first excited state wave function grows since its probability density largely locates within the same regions. At the same time, Figure 6c,d show the transition energy from the ground state to the first excited state, with and without a donor impurity. It can be observed that as the magnetic field increases, this quantity diminishes. This happens because the action of the magnetic field is to bring together the energies of the ground state and the first excited state for m=0 (see Figure 3). So, the higher the magnetic field, the smaller the difference between these energies. From both plots, it can be seen that the transition energy is higher for small magnetic fields with donor impurity effects than when the impurity is not considered. This is due to the impurity-induced energy shift towards lower values, making the difference in $E_{b}$ greater. This shift has an average magnitude of 0.412 meV.

Figure 7 concerns to the same kind of study reported in previous Figure 5 and Figure 6, considering the applied electric field as the external probe. Results plotted in Figure 7a show that as the applied electric field strengthens, there is a magnitude decay of $|M_{12}{|}^{2}$. Initially. it reaches a maximum value for $F_{z}=0$, and then, it decreases. This maximum appears because the ground state wave function is uniformly distributed over the QD confinement region. In contrast, the first excited state has its lobes concentrated at $z=\pm R_{2}$, thus producing the largest wave function overlap. The decreasing nature of $|M_{12}{|}^{2}$ is because the electric field pushes |1〉 and |2〉 toward $z=-R_{2}$; for example, at $F_{z}=50$ kV/cm the state |1〉 is concentrated at $z=-R_{2}$, and the lower lobes of state |2〉 are also concentrated there, giving the lowest overlap (see inset in Figure 7). In Figure 7b, the results presented include the effects of the donor impurity. A behavior very similar to the system without impurity is noted, with the difference that, for small values of the electric field, $|M_{12}{|}^{2}$ has a smaller magnitude. For example, for $F_{z}=0$, without impurities, $|M_{12}{|}^{2}$ has a value of 19.024 nm${}^{2}$, and 18.208 nm${}^{2}$ with impurity, although this difference begins to disappear as the electric field increases. This situation has to do with the electron-impurity interaction. For low electric fields, it is the dominant one -since the electron is closer to the impurity-. In contrast, it loses strength in large fields due to the more significant electron-impurity separation. In this case, the ground state wave function gets closer to the impurity, so the overlap is smaller than in Figure 7a. Finally, Figure 7c,d show the transition energy from the ground state to the first excited state, with and without impurity in the structure. In this plot, $E_{12}$ increases as the electric field strengthens. The reason for this variation is that an increase in the electric field implies greater electron confinement. As plotted in Figure 4, all energies exhibit a decreasing trend, but the ground level decreases at a faster rate compared to the first excited one. When evaluating $E_{12}$, a more significant difference is obtained. On the other hand, by comparing Figure 7c,d, it is observed that for zero electric fields, the transition energy is greater for the case with impurity, but as the electric field increases, this difference disappears, which indicates that the action of electric field predominates in the system.

The results reported in Figure 5, Figure 6 and Figure 7 for the transition energies and $|M_{12}{|}^{2}$ factor are the key elements to analyze the features of the optical responses of interest in this work. That is, total optical absorption coefficient (α) and relative changes in the total refractive index coefficient (Δn), both as functions of the incident photon energy [through Equations (Equation 7) and (Equation 10)]. Accordingly, the incoming electromagnetic wave is assumed to be polarized along the *z*-direction. Results will be presented considering both the absence and presence of the donor impurity at the center of the QD. In Figure 8, three values of the internal radius ($R_{1}$) are considered, taking into account the results of Figure 5. In Figure 9, three values of the applied magnetic field will be shown, taking into account the results of Figure 6. In Figure 10, three values of the applied electric field will be shown, taking into account the results of Figure 7. The light intensity value used for the calculations shown below is 30 MW/cm${}^{2}$.

In Figure 8 the reported results correspond to the optical absorption coefficient and the relative refractive index change coefficient, both as functions of the incident photon energy in a ZnS/CdS/ZnS spherical core/shell/shell QD for several values of the internal dot radius. Figure 8a,b show that there is a drop in the resonant peak magnitude for the linear absorption coefficient ($\alpha^{\left(1\right)}$) as the inner radius increases. From Equation (Equation 5) it is seen that the $\alpha^{\left(1\right)}$ maximum amplitude is related to the quantity $E_{12}{\left|M_{12}\right|}^{2}$. According to our results, $|M_{12}{|}^{2}$ increases (see Figure 5a,b) and ($E_{12}$) decreases (see Figure 5c,d) due to the greater degree of confinement. In both cases, as a result, the $\alpha^{\left(1\right)}$ amplitude decreases as the radius increases. This occurs because the $E_{12}$ value decays in greater proportion than the increase of $|M_{12}{|}^{2}$. Accordingly, the decrease in amplitude, and the peak shift of $\alpha^{\left(1\right)}$ to the red, are directly related to the $E_{12}$ behavior. The opposite effect occurs for the third-order absorption coefficient ($\alpha^{\left(3\right)}$) since it shows an increase in amplitude as the radius increases. This behavior is explained by Equation (Equation 6), in which it is observed that the $\alpha^{\left(3\right)}$ maximum amplitude is related to $E_{12}{\left|M_{12}\right|}^{4}$. So, the fourth power dominates the increase in the product. With Equation (Equation 7) and the above in mind, the behavior of the total absorption coefficient (α) is explained. This quantity decreases both in its amplitude and in the position of the peaks (redshift) due to the increase of the inner radius. The fall in the amplitude of the total absorption coefficient is further enhanced by the increase in the magnitude of $\alpha^{\left(3\right)}$. Analyzing α in the presence of the on-center donor impurity allows observing that, for a small radius, α exhibits a greater amplitude and a shift to higher energy as compared to the no-impurity case; this is because $E_{12}$ has a higher value for this case (see Figure 5c,d). Besides, Figure 8c,d show the linear and third-order contributions to the relative change in the total refractive index coefficient. It is observed that both $\Delta n^{\left(1\right)}$ and $\Delta n^{\left(3\right)}$ increase in amplitude and shift to the red part of the spectrum as the internal radius increases, with or without impurities. This behavior is driven by the increase of $|M_{12}{|}^{2}$ and the decrease of ($E_{12}$), Equations (Equation 8) and (Equation 9), producing very similar results in both cases with and without impurity.

The results depicted in Figure 9 are as those appearing in Figure 8, but with the consideration of the influence of an applied magnetic field with increasing strength. Figure 9a,b present the peak magnitude and energy of the total, linear, and non-linear optical absorption coefficients. It is possible to notice that these coefficients decrease in magnitude and energy position as long as the magnetic field increases. This diminishing trend relates to the reduction in the value of the $E_{12}{\left|M_{12}\right|}^{2}$ term as the magnetic field increases (see Figure 6a,b). The energy shift towards the red results from the decreasing behavior of $E_{12}$ as the magnetic field intensity augments. A reflection of the rise in $|M_{12}{|}^{2}$ (see Figure 9c,d) are the changes suffered by Δn, $\Delta n^{\left(1\right)}$, and $\Delta n^{\left(3\right)}$ with the increase of the magnetic field. Their peaks increase in amplitude, in all cases, with and without the effect of impurity. Again, the energy shift to lower energies is due to the decreasing character of $E_{12}$. When comparing the results with and without impurity, there are no very appreciable changes in the magnitude and position on the energy scale, which reaffirms what is shown in Figure 6, that the impurity effect on the electron is small.

The calculated optical coefficients shown in Figure 10 follow the same structure as in Figure 8 and Figure 9 but were evaluated for several values of an applied electric field. Figure 10a,b show a clear decrease in the magnitude of both the absorption coefficient peak and the relative refraction index change coefficient as the intensity of the applied electric field increases. This occurs because, unlike the previous results in Figure 8 and Figure 9, $|M_{12}{|}^{2}$ strongly decreases whilst $E_{12}$ strongly increases with the field [see Figure 7]. The product $E_{12}{\left|M_{12}\right|}^{2}$ decreases as the electric field intensifies. A shift in the energy values towards the blue appears because $E_{12}$ increases as the electric field increases [see Figure 7c,d]. Finally, in Figure 10c,d, it is shown that Δn decreases as the electric field increases and shifts to higher values of the incident photon energy due to the same reasons discussed above. An important result to keep in mind is that the impurity interacts more with the electron for zero electric fields, but as the field increases, the influence of the impurity becomes almost null. It is worth mentioning that $\alpha^{\left(3\right)}$, and $\Delta n^{\left(3\right)}$ are not shown in the graphs because depending on $|M_{12}{|}^{4}$, their amplitudes are very small, and their contribution to the total values of α, and Δn, is unimportant.

Finally, it is worth commenting that the so-called colloidal quantum dots correspond to the type of multilayer structures reported in this study. In general, their sizes are in the range of 4 nm and 10 nm in diameter, and their optical properties for technological uses are modifiable with electric fields in the range of several tens of kV/cm and magnetic fields in the range from 0–50 T [42,43,44]. We hope that this study will serve as motivation for researchers in the experimental field to undertake the task of synthesizing the type of multilayer structures such as the one reported in this article and to consider the possibility of technological applications.

## 4. Conclusions

We have investigated the nonlinear optical response of ZnS/CdS/ZnS core/shell/shell spherical quantum dots related to energy transitions between the ground state (|1〉) and the first excited state (|2〉) in the system. For that purpose, the electronic states were determined from the solution of the effective mass equation -with and without the presence of a donor impurity (which was assumed to be placed at the center of the spherical complex)- using the finite element method as implemented in COMSOL-Multiphysics software. The study considers the influence of changes in dot size and the presence of static electric and magnetic fields externally applied to the system. It is determined that geometrical modifications, as well as the increment in applied field intensities, have a noticeable impact on the amplitude and the position of total -linear plus third-order nonlinear- light absorption and relative refractive index change coefficients. Besides, the inclusion of impurity effects reveals significant modifications in the electronic spectrum, which also reflect on the investigated optical properties.

Results obtained for peak energies and amplitudes in the optical coefficients could be used in the designing of optoelectronic devices, which would be tuned to different energies and amplitudes by changing the external probes and electric and magnetic fields. For instance, in this case, transition energies fall within the range 1.5–20 meV, which corresponds to the terahertz region.

## Figures and Tables

**Figure 1 nanomaterials-13-00550-f001:**
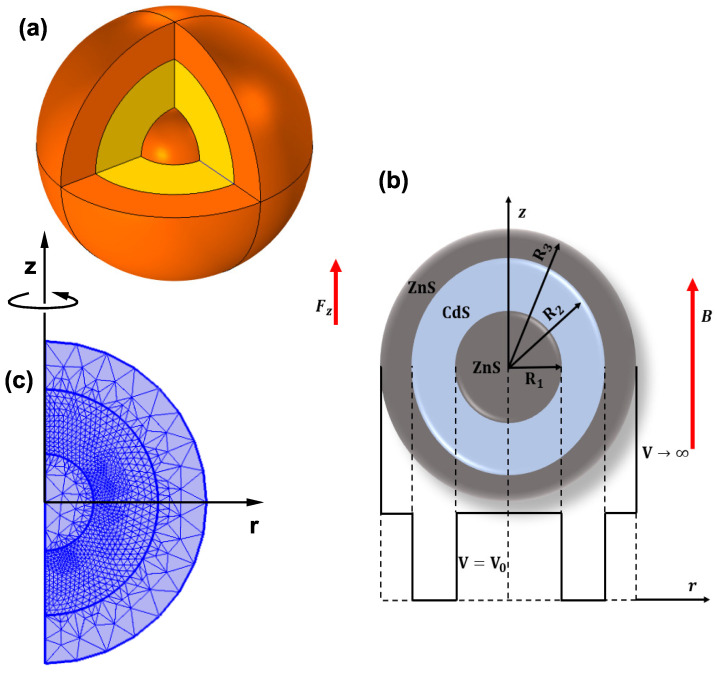
(**a**) ZnS/CdS/ZnS spherical quantum dot heterostructure considered in this work. In (**b**), the dimensions of the quantum dot and the applied axial electric and magnetic field are indicated together with the radial dependent confinement potential. In (**c**) is depicted the φ=0-projection of the heterostructure (where cylindrical coordinates are considered) together with the refined mesh used in the finite element method calculations.

**Figure 2 nanomaterials-13-00550-f002:**
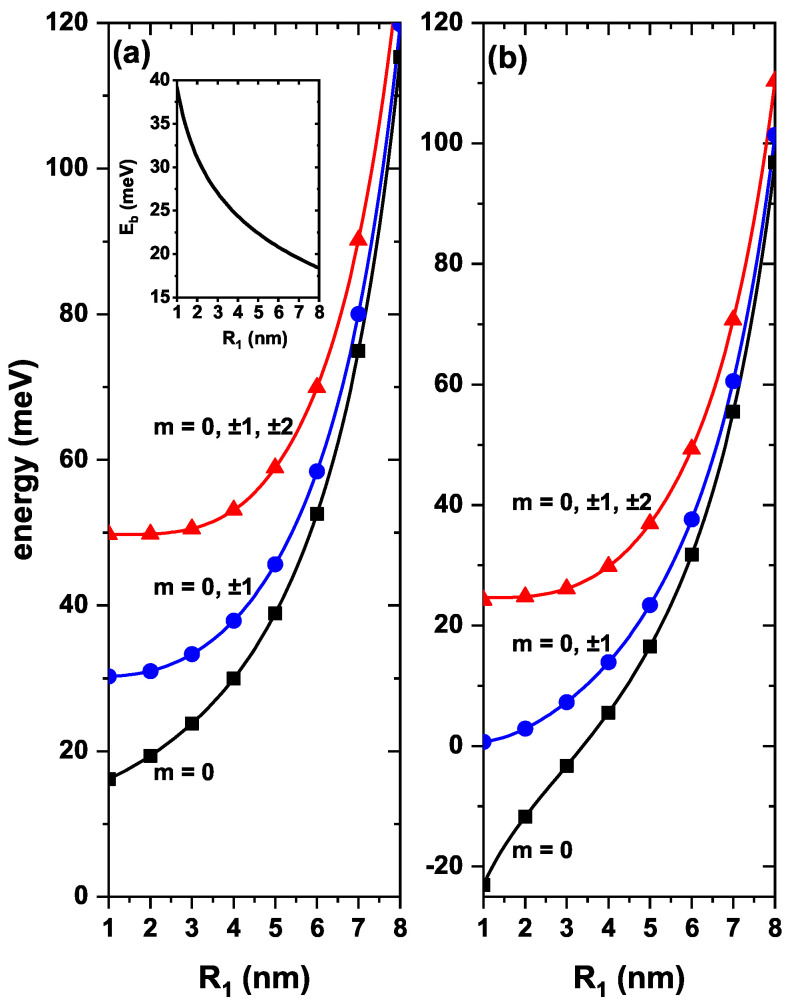
The first three lowest electronic states of a ZnS/CdS/ZnS spherical core/shell/shell quantum dot as functions of the $R_{1}$ radius. Figure (**a**) presents the results without, and (**b**) those with effects of an on-center donor impurity, with $R_{2}=11$ nm and $R_{3}=12$ nm, for different values of the *m*-quantum number and without effects of external electric and magnetic fields. Labels close to the curves indicate the values of the *m*-quantum number. The inset in Figure (**a**) corresponds to the ground state (lowest state with m=0) binding energy between the impurity and the electron as a function of the internal dot radius. In both panels, solid symbols correspond to the energies obtained by solving the three-dimensional problem using Equation (Equation 1).

**Figure 3 nanomaterials-13-00550-f003:**
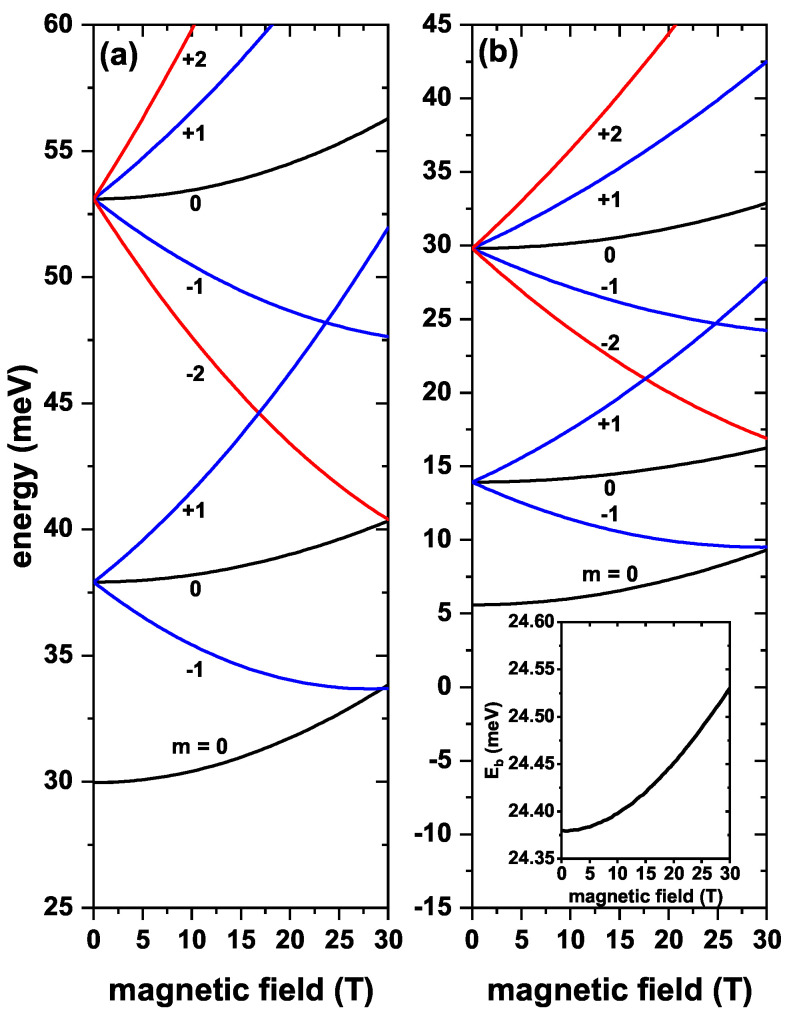
The lowest three electronic states in a ZnS/CdS/ZnS spherical core/shell/shell quantum dot as functions of the applied magnetic field. Plot (**a**) contains results without, and (**b**) includes on-center donor impurity effect, for $R_{1}=4$ nm, $R_{2}=11$ nm, and $R_{3}=12$ nm. Different values of the *m*-quantum number and without external electric field effects have been considered. Labels close to the curves give the values of the *m*-quantum number. The inset in (**b**) corresponds to the ground state (lowest state with m=0) binding energy as a function of the applied magnetic field.

**Figure 4 nanomaterials-13-00550-f004:**
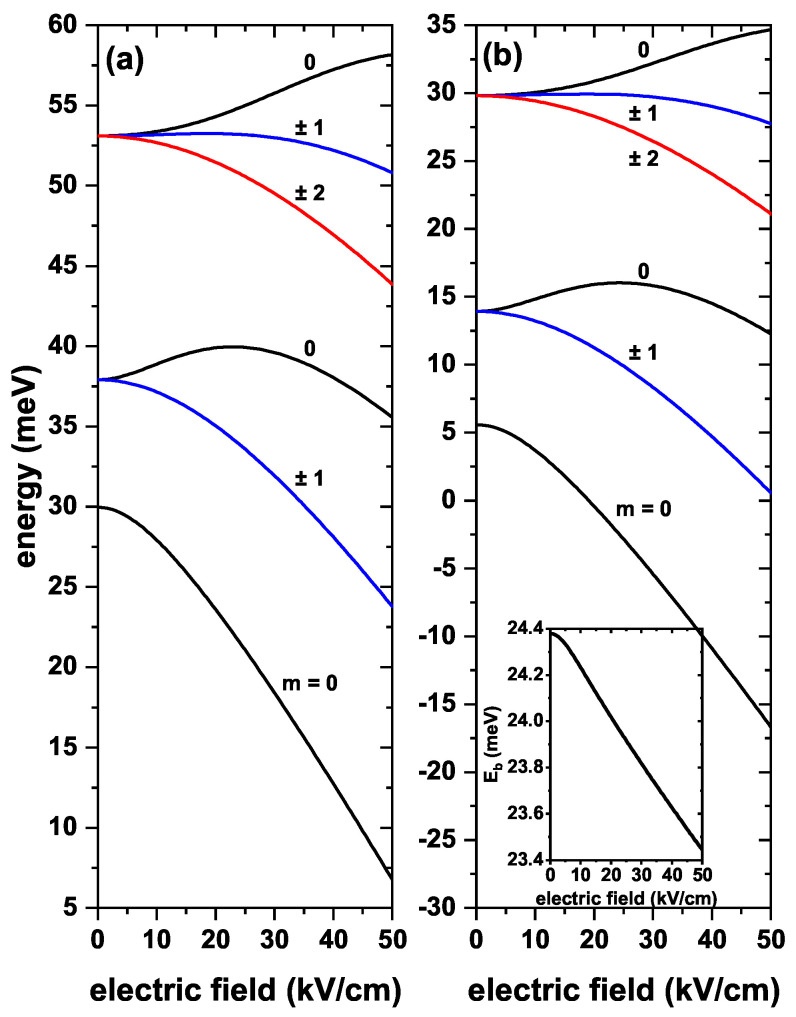
The lowest electronic states in a ZnS/CdS/ZnS spherical core/shell/shell quantum dot as functions of the applied electric field. In (**a**), results are presented without, and in (**b**) with on-center donor impurity effects, with $R_{1}=4$ nm, $R_{2}=11$ nm, and $R_{3}=12$ nm. Different values of the *m*-quantum number and without external magnetic field effects have been considered. Labels above each of the curves give the values of the *m*-quantum number. The inset in (**b**) corresponds to the ground state (the lowest state with m=0) binding energy as a function of the applied electric field.

**Figure 5 nanomaterials-13-00550-f005:**
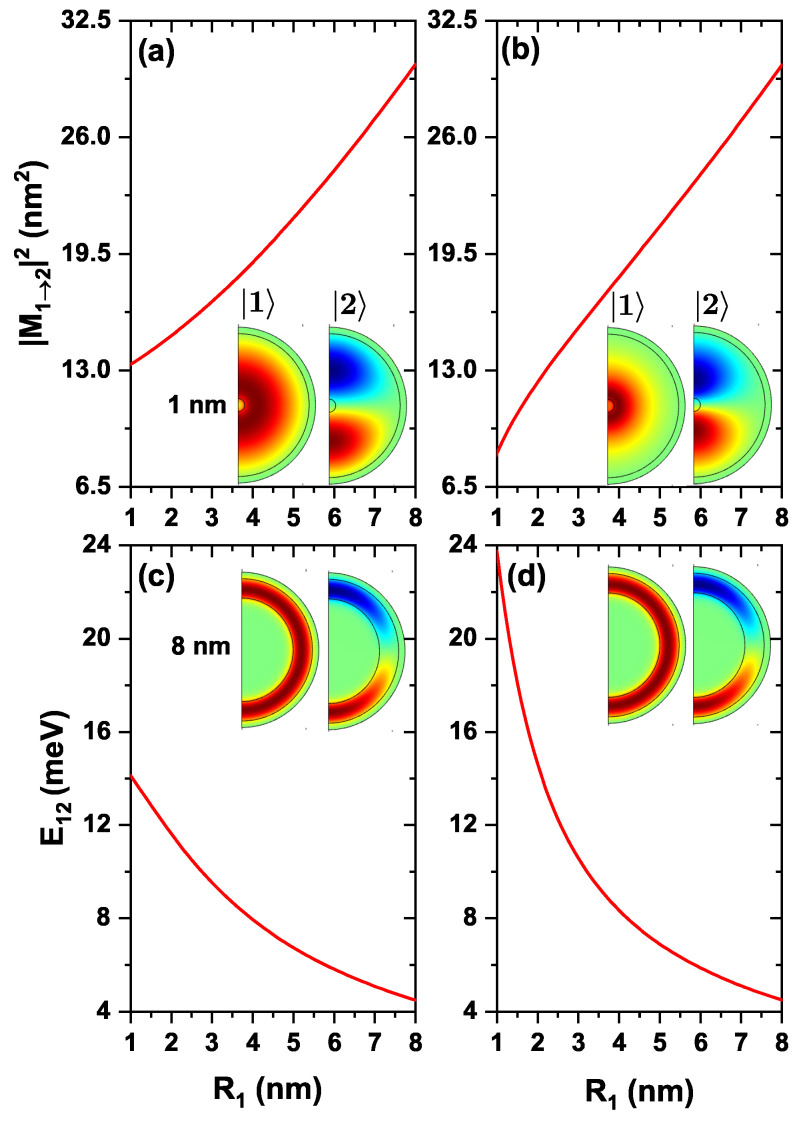
The variation of the $|M_{12}{|}^{2}$ (**a**,**b**), and the transition energy (**c**,**d**), as functions of the internal radius $R_{1}$, for transitions from the ground state (|1〉) to the first excited state (|2〉) with m=0, for an electron confined in a ZnS/CdS/ZnS spherical QD. (**a**,**c**), present the results without, and (**b**,**d**) with the effects of a donor impurity located at the center of the dot system. Calculations are with $R_{2}=11$ nm and $R_{3}=12$ nm, without effects of external electric and magnetic fields.

**Figure 6 nanomaterials-13-00550-f006:**
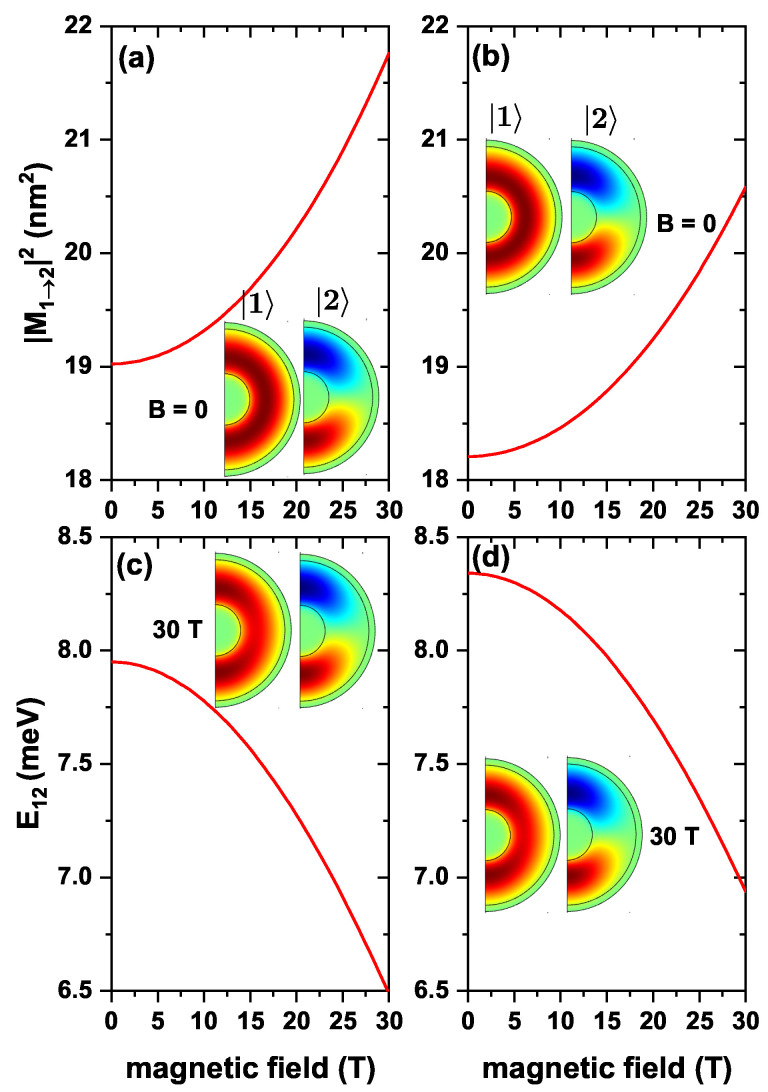
The variation of $|M_{12}{|}^{2}$ (**a**,**b**), and the transition energy (**c**,**d**), as a function of the magnetic field, for transitions from the ground state (|1〉) to the first excited state (|2〉) with m=0, for an electron confined in a ZnS/CdS/ZnS spherical QD. (**a**,**c**) are without, and (**b**,**d**) are with the effects of a donor impurity located at the QD center. Calculations are with $R_{1}=4$ nm, $R_{2}=11$ nm, and $R_{3}=12$ nm, without electric field effects.

**Figure 7 nanomaterials-13-00550-f007:**
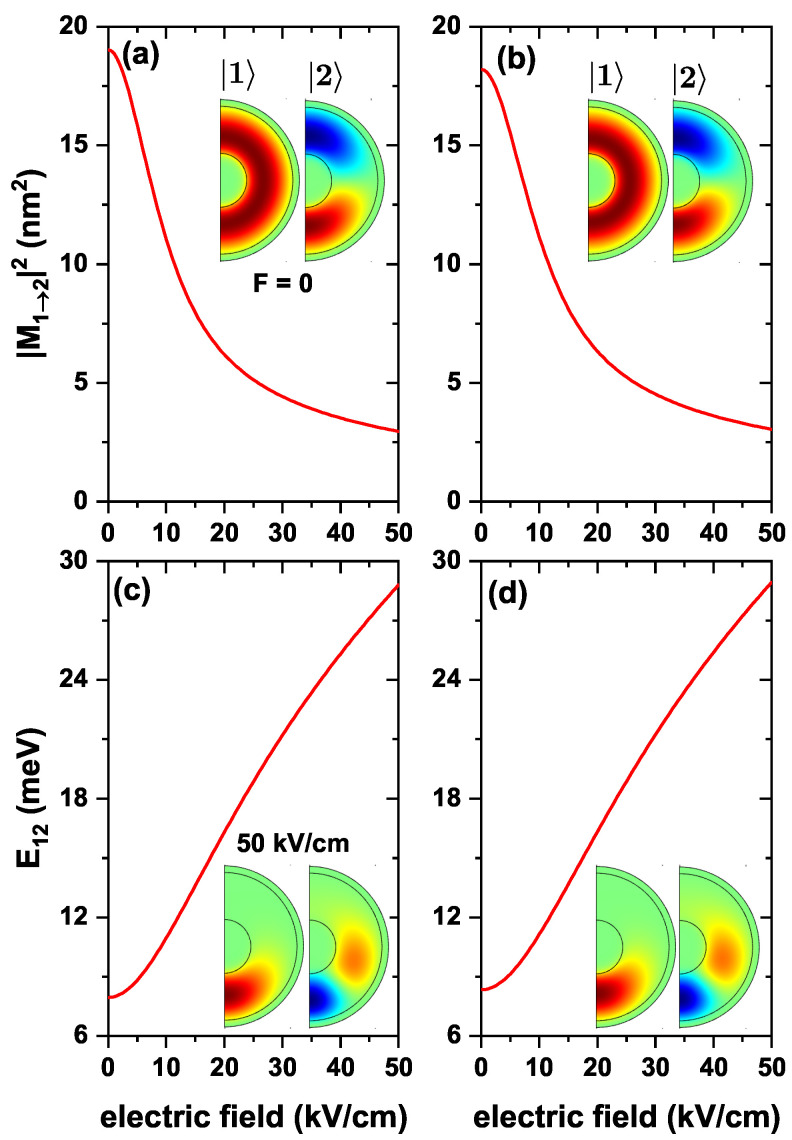
The variation of the $|M_{12}{|}^{2}$ (**a**,**b**), and the transition energy (**c**,**d**), as a function of the electric field $F_{z}$, for transitions from the ground state (|1〉) to the first excited state (|2〉) with m=0, for an electron confined in a ZnS/CdS/ZnS spherical QD. (**a**,**c**) results without, and (**b**,**d**) with the effects of a donor impurity, located at the QD center, with $R_{1}=4$ nm, $R_{2}=11$ nm and $R_{3}=12$ nm, without external magnetic field effect.

**Figure 8 nanomaterials-13-00550-f008:**
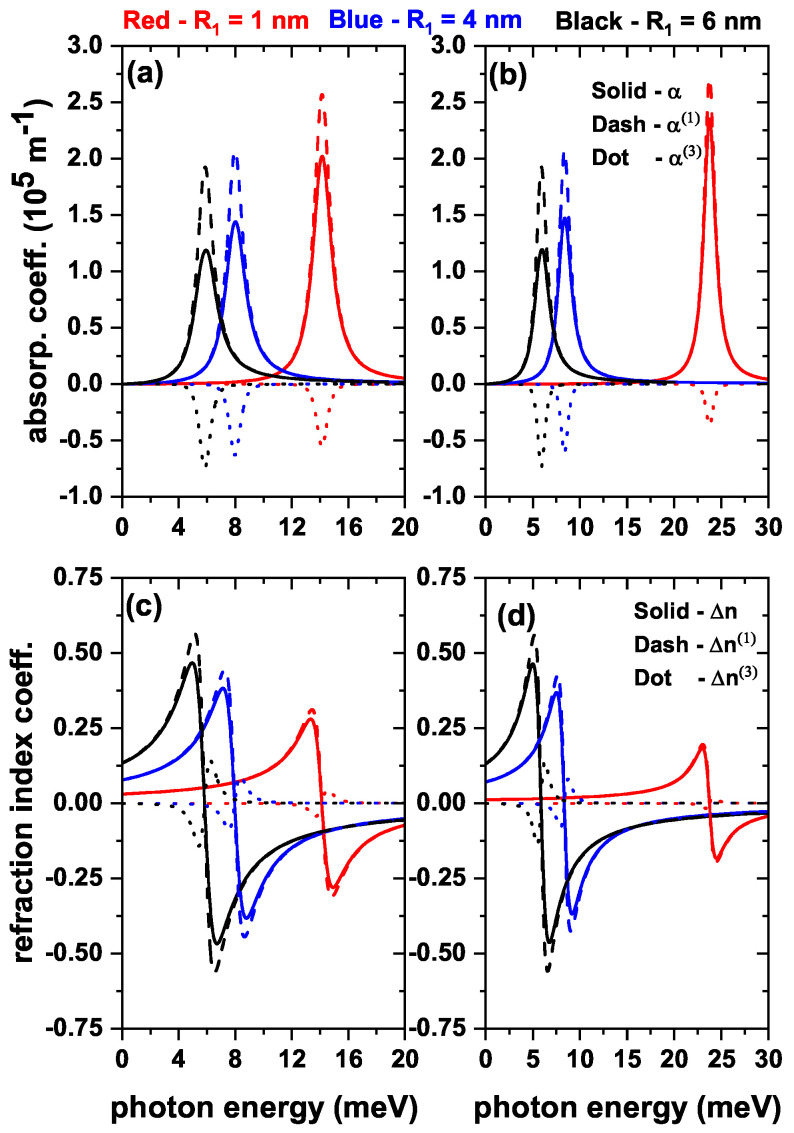
The total (solid line), linear (dashed line), and nonlinear (dotted line) optical absorption coefficient (**a**,**b**), and the total (solid line), linear (dashed line), and nonlinear (dotted line) relative changes in the refractive index coefficient (**c**,**d**) as functions of the incident photon energy in a ZnS/CdS/ZnS spherical core/shell/shell quantum dot, for three values of the internal radius: 1 nm (red line), 4 nm (blue line), and 6 nm (black line). Results are without (**a**,**c**) and with (**b**,**d**) donor impurity at the center of the dot. Calculations are with $R_{2}=11$ nm and $R_{3}=12$ nm. The optical transition is between the states |1〉→|2〉, with m=0 and without external electric and magnetic field effects.

**Figure 9 nanomaterials-13-00550-f009:**
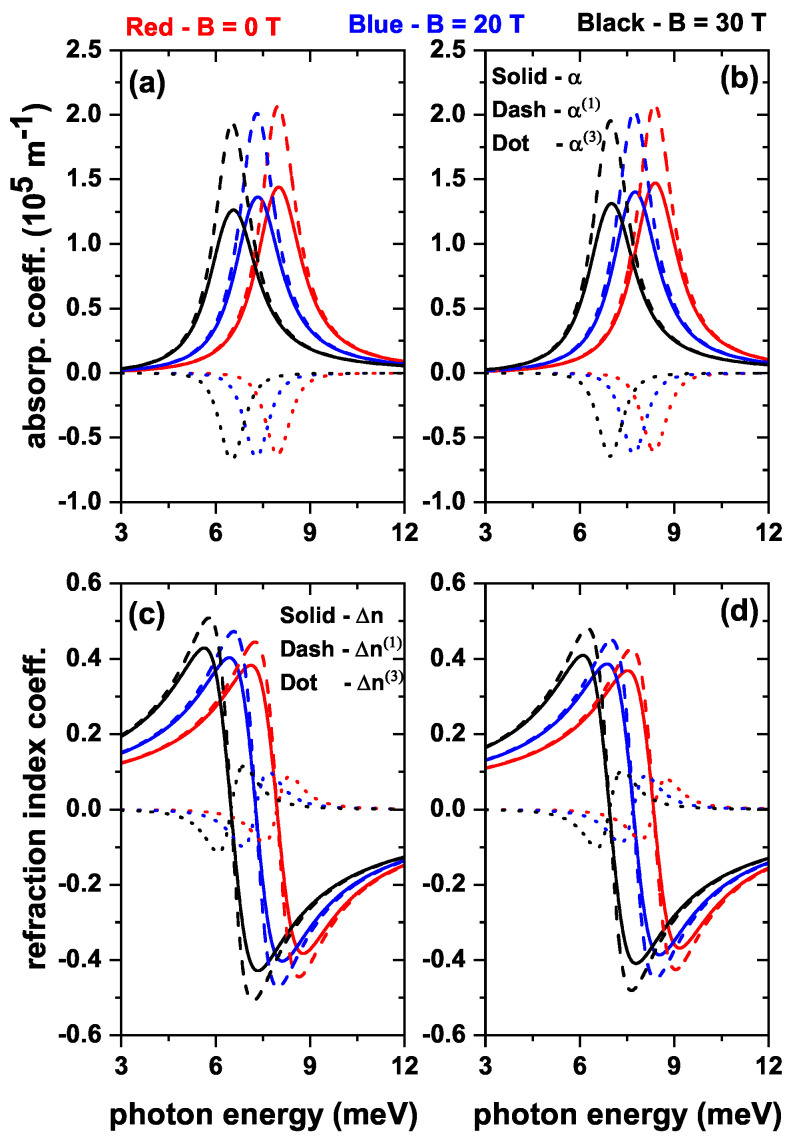
The total (solid line), linear (dashed line), and nonlinear (dotted line) optical absorption coefficient (**a**,**b**), and the total (solid line), linear (dashed line), and nonlinear (dotted line) relative changes in the refractive index coefficient (**c**,**d**) as functions of the incident photon energy in a ZnS/CdS/ZnS spherical core/shell/shell quantum dot, for three values of the magnetic field: zero (red line), 20 T (blue line), and 30 T (black line). The results are without (**a**,**c**) and with (**b**,**d**) donor impurity at the center in the QD. Calculations are with $R_{1}=4$ nm, $R_{2}=11$ nm, and $R_{3}=12$ nm. The optical transition is between the states |1〉→|2〉, with m=0 and without external electric field effects.

**Figure 10 nanomaterials-13-00550-f010:**
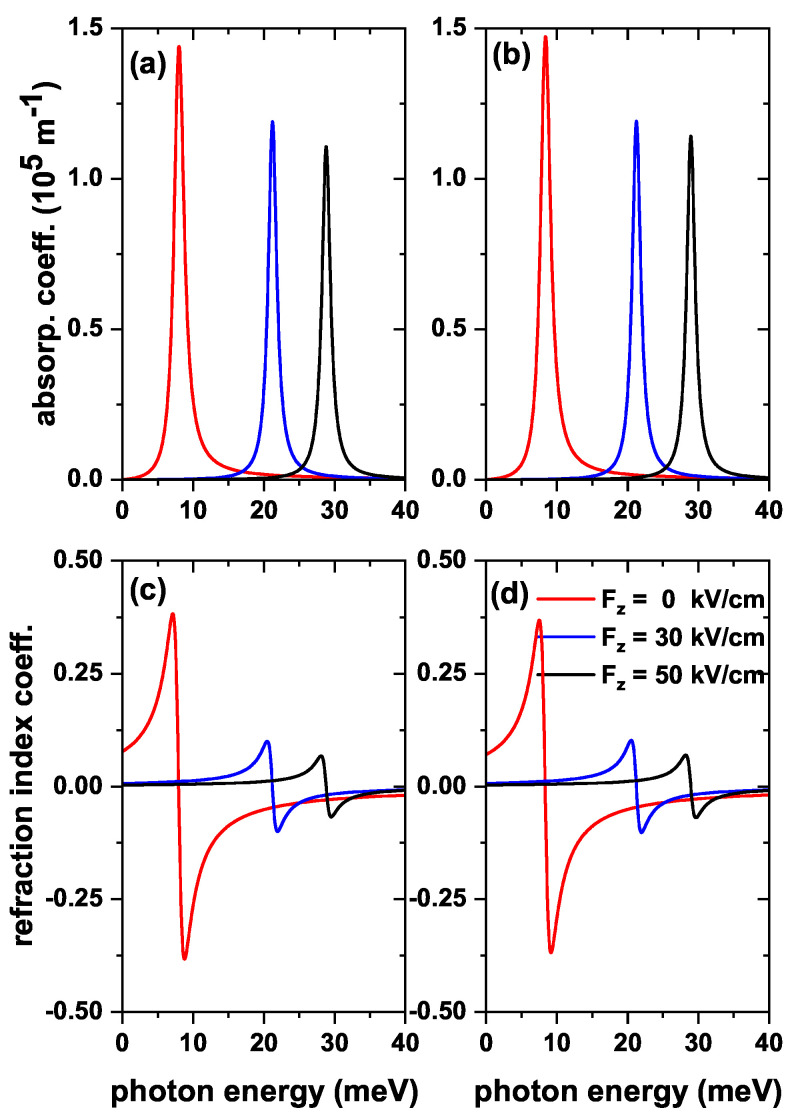
The total optical absorption coefficient (**a**,**b**) and relative changes in the total refractive index coefficient (**c**,**d**) as a function of the energy of the incident photon in a ZnS/CdS/ZnS spherical QD, for three values of the electric field: zero, 30 kV/cm, and 50 kV/cm. The results are without (**a**,**c**) and with (**b**,**d**) donor impurity at the QD center. Calculations are for $R_{1}=4$ nm, $R_{2}=11$ nm, and $R_{3}=12$ nm. The optical transition is between the states |1〉→|2〉, with m=0 and without external magnetic field effects.

## Data Availability

No new data were created or analyzed in this study. Data sharing is not applicable to this article.

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
