# Peer review of "Optical Properties in a ZnS/CdS/ZnS Core/Shell/Shell Spherical Quantum Dot: Electric and Magnetic Field and Donor Impurity Effects"

_nanomaterials, 2023, doi:10.3390/nano13030550_

Round 1

Reviewer 1 Report

This paper is to analyze about optical properties in a ZnS/CdS/ZnS core/shell/shell spherical quantum dot by using the effective mass approximation in COMSOL-Multiphysics software calculation. There are some interesting optical property results discussed in this paper. However, some points are suggested to the authors as following.

1.     The authors applied the axis-symmetric module to calculate the optical properties due to the QD is a sphere. Could the proposal calculate the optical properties of the QD in a non-axis-symmetric module? What are the difficulties when using non-axis-symmetric module?

2.     How do the authors verify that the analysis model in calculation is corrected? The authors are suggested to give some simple real cases for supporting their analysis model.  

3.     Some figures show unclear information inside, for example, Figure 2 shows the number 1, 3, and (1) m = 0 (2) m = -1, 0, 1, …. The authors are suggested to explain those in more details.  

What could be the best parameters for its application? And is it possible to be fabricated through current semi-conductor technology?

Author Response

Report 1

The Referee:

This paper is to analyze about optical properties in a ZnS/CdS/ZnS core/shell/shell spherical quantum dot by using the effective mass approximation in COMSOL-Multiphysics software calculation. There are some interesting optical property results discussed in this paper. However, some points are suggested to the authors as following.

Our reply:

We want to thank the Referee's observations and comments, which have allowed us to substantially improve the quality of our article.

The Referee:

  1. The authors applied the axis-symmetric module to calculate the optical properties due to the QD is a sphere. Could the proposal calculate the optical properties of the QD in a non-axis-symmetric module? What are the difficulties when using non-axis-symmetric module?

Our reply:

Actually, the axis-symmetric module is used to determine the electronic properties (i.e. energy values and corresponding wave functions) of the QD. It is not directly used the calculate the optical properties. Yet optical properties are determined by using the energy values and wave functions in Eqs. 5-10. On the other hand, the system has a symmetry along the azimuthal axis. Of course the non-axis symmetric module can be used to calculate the electronic structure of the system considered, but using of the non-axis-symmetric module needs more computational time and effort. Therefore, in calculations, using the non-axis symmetric module may not be so meaningful for this QD structure.

The Referee:

  1. How do the authors verify that the analysis model in calculation is corrected? The authors are suggested to give some simple real cases for supporting their analysis model.

Our reply:

In the second paragraph of the discussion of Fig. 2 we have added the following text:

For purposes of confirming the results, we have proceeded to obtain the solution directly from Eq. (1), which corresponds to a three-dimensional problem. The results obtained are reported in Fig. 2 with solid symbols. As can be seen, the results coincide exactly for both the 3D-solution, Eq. (1), and for the 2D-solution, Eq. (4), which uses the axial symmetry of the problem. Likewise, in the case of the problem without impurity and the absence of electric and magnetic fields, the problem of an electron confined in a multilayer QD with spherical symmetry has a quasi-exact solution where the wave function in the different regions of the QD can be written as a linear combination of radial Bessel functions. Applying the Ben Daniel-Duke boundary conditions, we arrive at a transcendental equation whose numerical solution leads to obtaining the energies of the confined particle. The full symbols have also been obtained by solving this quasi-exact problem, again showing an excellent agreement for the three calculation methods used in this study. One of the differences between obtaining the solution of Eqs. (1) and (4) lie in the computation time. Using a 2.8 Ghz Intel i7-processor, obtaining the ground state energy for a single value of $R_1$ in Fig. 2(a), using Eq. (4), it takes 3\,s of computing time, a situation that increases by a factor of 4 to obtain the same energy in the 3D-module that corresponds to the solution of Eq. (1).

The Referee:

  1. Some figures show unclear information inside, for example, Figure 2 shows the number 1, 3, and (1) m = 0 (2) m = -1, 0, 1, …. The authors are suggested to explain those in more details.

Our reply:

We thank the Referee for his observation. We have improved the presentation of Figs. 2, 3, and 4 to clarify the labels on each curve which correspond to the m-quantum number.

The Referee:

  1. What could be the best parameters for its application? And is it possible to be fabricated through current semi-conductor technology?

Our reply:

We want to thank the Referee for his/her comment and questions. In this study we present a combination of materials in wells and barriers that gives rise to spherical quantum dots that can be obtained technologically and are known as Colloidal Quatum Dots. Development in this field has been intense in the last decade with multiple applications in different areas of science. In the final part of the Conclusions section we have added the following paragraph with its corresponding references.

Colloidal quantum dots correspond to the type of multilayer structures reported in this study. In general, their sizes are in the range of 4\,nm and 10\,nm in diameter and their optical properties for technological uses are modifiable with electric fields in the range of several tens of kV/cm and magnetic fields in the range from 0-50\,T \cite{39,40,41}. We hope that this study will serve as motivation for other researchers in the experimental field to synthesize the type of multilayer structures such as the one reported in this article and to consider the possibility of technological applications.

Finally, we hope that the Referee finds our responses and changes to the manuscript satisfactory. We hope that our article in its revised version will be suitable for publication in the journal Nanomaterials.

Reviewer 2 Report

In this paper, the relationship between the nonlinear optical response of ZnS/CdS/ZnS core/shell/shell spherical quantum dots and the energy transition between the ground state (|1⟩) and the first excited state (|2⟩) in the system is studied. Considering the influence of dot size change and the existence of external static electricity and magnetic field applied to the system, this research is a meaningful work. Meanwhile, the addition of impurity effect shows a significant change in the electronic spectrum, which is also a good supplement to the optical properties studied. I think it is suitable for publication in Nanomaterials journal following some minor revisions as follows.

1.      There are more identical sentence patterns. For example, line 127, line 167, and line 196,···are depicted the···”. Please rewrite the manuscript.

2.      Please specify. For example, P12, line 279, Which pictures do "the above Figures" refer to?

3.      There are some repeated statements in this manuscript. For example, line 291-293, line 319-321, and line 335-337, “···are depicted the results for the optical absorption coefficient and the relative changes in the refractive index coefficient as functions of the incident photon energy in a ZnS/CdS/ZnS spherical core/shell/shell QD for several values of···”.

4.      The formats of writing the reference are not uniform. The authors are advised to carefully checked and corrected.

Such as: ref. 38, Change “Donor Impurity in CdS/ZnS Spherical Quantum Dots under Applied Electric and Magnetic Fields” to “Donor impurity in CdS/ZnS spherical quantum dots under applied electric and magnetic fields”.

Overall, the whole manuscript is not well written and some errors need be revised.

Author Response

Report 2

The Referee:

In this paper, the relationship between the nonlinear optical response of ZnS/CdS/ZnS core/shell/shell spherical quantum dots and the energy transition between the ground state (|1⟩) and the first excited state (|2⟩) in the system is studied. Considering the influence of dot size change and the existence of external static electricity and magnetic field applied to the system, this research is a meaningful work. Meanwhile, the addition of impurity effect shows a significant change in the electronic spectrum, which is also a good supplement to the optical properties studied. I think it is suitable for publication in Nanomaterials journal following some minor revisions as follows.

Our reply:

We want to thank the Referee's observations and comments, which have allowed us to substantially improve the quality of our article.

The Referee:

  1. There are more identical sentence patterns. For example, line 127, line 167, and line 196, “···are depicted the···”. Please rewrite the manuscript.

Our reply:

We thank the Referee for his/her very kind observation. We have carefully revised the manuscript and corrected the repetitive presentation of many of the introductory phrases of each figure.

The Referee:

  1. Please specify. For example, P12, line 279, Which pictures do "the above Figures" refer to?

Our reply:

We want to thank the Referee for his/her very kind observation. We have improved the presentation of the introductory phrase from which the discussion in Figs. 8, 9, and 10 follows. In the revised version of the manuscript, the phrase has remained as follows:

The results reported in Figs. 5, 6, and 7 for the transition energies and the $|M_{1-2}|^2$-parameters are the key elements to analyze the features of the optical responses of interest in this work.

The Referee:

  1. There are some repeated statements in this manuscript. For example, line 291-293, line 319-321, and line 335-337, “···are depicted the results for the optical absorption coefficient and the relative changes in the refractive index coefficient as functions of the incident photon energy in a ZnS/CdS/ZnS spherical core/shell/shell QD for several values of···”.

Our reply:

We thank the Referee for his/her very kind observation. We have carefully revised the manuscript and corrected the repetitive presentation of many of the introductory phrases of each figure.

The Referee:

  1. The formats of writing the reference are not uniform. The authors are advised to carefully checked and corrected.

Such as: ref. 38, Change “Donor Impurity in CdS/ZnS Spherical Quantum Dots under Applied Electric and Magnetic Fields” to “Donor impurity in CdS/ZnS spherical quantum dots under applied electric and magnetic fields”.

Our reply:

We thank the Referee for his observation. We have reviewed and written all references in the Nanomaterials Journal format.

The Referee:

Overall, the whole manuscript is not well written and some errors need be revised.

Our reply:

The manuscript has been carefully reviewed. Multiple spelling and grammar errors have been corrected.

Finally, we hope that the Referee finds our responses and changes to the manuscript satisfactory. We hope that our article in its revised version will be suitable for publication in the journal Nanomaterials.
